# Current Therapeutic Strategies in Diabetic Foot Ulcers

**DOI:** 10.3390/medicina55110714

**Published:** 2019-10-25

**Authors:** Aurelio Perez-Favila, Margarita L Martinez-Fierro, Jessica G Rodriguez-Lazalde, Miguel A Cid-Baez, Michelle de J Zamudio-Osuna, Ma. del Rosario Martinez-Blanco, Fabiana E Mollinedo-Montaño, Iram P Rodriguez-Sanchez, Rodrigo Castañeda-Miranda, Idalia Garza-Veloz

**Affiliations:** 1Molecular Medicine Laboratory, Unidad Academica de Medicina Humana y Ciencias de la Salud, Universidad Autonoma de Zacatecas, Carretera Zacatecas-Guadalajara Km.6. Ejido la Escondida, C.P., Zacatecas 98160, Mexico; chaure7@gmail.com (A.P.-F.); rodriguezlazaldej@yahoo.com (J.G.R.-L.); drcidbaez@uaz.edu.mx (M.A.C.-B.); 2Posgrado en Ingenieria y Tecnologia Aplicada, Unidad Academica de Ingenieria Electrica, Universidad Autonoma de Zacatecas, Av. Ramon Lopez Velarde No. 801, Col. Centro, C.P., Zacatecas 98000, Mexicorcastm@gmail.com (R.C.-M.); 3Bioss Celulas Madre, Av. Gomez Morin No. 1200, Carrizalejo, C.P., San Pedro Garza Garcia 66254, Nuevo Leon, Mexico; michellezamudiosuna@gmail.com; 4Unidad Academica de Enfermeria, Universidad Autonoma de Zacatecas, Carretera Zacatecas-Guadalajara Km.6. Ejido la Escondida, C.P., Zacatecas 98160, Mexico; fabianamollinedo@yahoo.com; 5Laboratorio de Fisiologia Molecular y Estructural, Facultad de Ciencias Biologicas, Universidad Autonoma de Nuevo Leon, C.P., San Nicolas de los Garza 66455, Nuevo Leon, Mexico; iramrodriguez@gmail.com

**Keywords:** therapy, tissue engineering, diabetic foot ulcers

## Abstract

Diabetic foot ulcers (DFUs) are the fastest growing chronic complication of diabetes mellitus, with more than 400 million people diagnosed globally, and the condition is responsible for lower extremity amputation in 85% of people affected, leading to high-cost hospital care and increased mortality risk. Neuropathy and peripheral arterial disease trigger deformities or trauma, and aggravating factors such as infection and edema are the etiological factors for the development of DFUs. DFUs require identifying the etiology and assessing the co-morbidities to provide the correct therapeutic approach, essential to reducing lower-extremity amputation risk. This review focuses on the current treatment strategies for DFUs with a special emphasis on tissue engineering techniques and regenerative medicine that collectively target all components of chronic wound pathology.

## 1. Introduction

Diabetes mellitus (DM) is a metabolic disorder characterized mainly by the presence of chronic hyperglycemia due to a deficiency of insulin secretion or insulin resistance [1]. The International Diabetes Federation (IDF) reported 425 million people globally with DM in 2017 [2].

Chronic complications of the disease vary according to the type of diabetes, time since onset, and degree of metabolic control, with the most prevalent being the following: neuropathy, 25%; retinopathy, 32%; and nephropathy, 23% [3]. The diabetic foot ulcer (DFU) is a complication of DM, with an annual incidence of 2.4–2.6% and a prevalence of 4–10% [4]. It is estimated that the rate of recurrence of DFUs is more than 50% after three years [5]. These complications constitute an increasing public health problem and are a leading cause of hospital ingress, amputation, and mortality in diabetes patients [6]. The economic impact of DFUs is significant. According to data from the IDF, USD 727 billion is spent on total DM health expenses in people aged 20–79 years [2].

Educating patients about podiatric care could prevent DFU development, and reduce their incidence. Understanding of the etiology of ulceration is essential to identify the tissue at risk of ulcerating and to enable proper preventive care, thereby reducing the incidence of foot lesions and, ultimately, amputations [7].

## 2. Etiology of Diabetic Foot Ulcers

For the appearance of a DFU, the convergence of several factors is necessary: usually, an initial injury (trauma) that is not detected by the patient because of an existing neuropathy, together with a peripheral vascular disease. The acquisition of an infection is a common origin of a DFU that can progress to a partial or complete lower limb amputation if not treated properly [8,9].

### 2.1. Diabetic Peripheral Neuropathy

Diabetic peripheral neuropathy (DPN) is defined as the presence of symptoms and/or signs of peripheral nerve dysfunction in people who have DM after excluding other causes [10]. DPN is the most common chronic complication in the lower limbs with a prevalence of over 60% in people with DM [11]. The effect on sensory, motor, and autonomic nerves can modify the ability of the patient to perceive certain stimuli such as pain, temperature, pressure, and touch [12]. Motor neuropathy can affect the small muscles of the foot causing atrophy; weakness; toe deformities; prominent metatarsals; and, in turn, limited joint mobility. On the other hand, autonomic neuropathy reduces sweating and increases temperature; combined with injuries that are not detected in time, limited mobility of joints, and deformity of the lower extremities associated with DPN, it can cause cracking of the skin, inflammation, and tissue necrosis, ultimately leading to the development of a DFU [9].

### 2.2. Peripheral Arterial Disease

Hyperglycemia together with oxidative stress produces the final products of advanced glycation, which are involved in the development of microvascular and macrovascular complications in people with DM [13]. Peripheral arterial disease (PAD) is a vascular condition characterized by atherosclerotic occlusive disease of the lower extremities that has been found in approximately 30% of patients with a DFU [14]. Their development is a gradual process, in which the artery becomes blocked, narrowed, or weakened, and, in addition to prolonged inflammation within the microcirculation, leads to thickening of the capillary, thus limiting the elasticity of capillaries leading to ischemia [9]. Related to atherosclerosis is media sclerosis Mönckeberg (MSM), which is frequent in type 2 DM, and is associated with progressive calcifications of the arterial wall media, leaving the intima intact [15]. MSM may interfere with compensatory remodeling of the arterial walls, and may accelerate the stenotic phase of the atherosclerosis [15,16]. PAD alone is not a cause of a DFU, but aggravates the damage caused by the combination of several risk factors such as DPN, foot deformities, skin dryness, trauma, and infection, enhancing the DFU development [8].

### 2.3. Infection

When a DFU appears, it is susceptible to the onset of infections, mainly owing to prolonged environmental exposure of the wound; pathogen-related factors such as density, virulence, and interactions; and immune defects linked to the host [9]. Several immunological defects have been reported in patients with DM, such as altered phagocytosis and bactericidal activity of polymorphonuclear cells; impaired chemotaxis and phagocytosis functions of monocyte/macrophage; disturbances of cellular innate immunity, including a low serum level of complement factor 4 (C4) and abnormal production of cytokines by monocytes [17]; and alteration of lymphocytes subpopulations and immunoglobulins levels [18]. These abnormalities, mostly related to innate immunity, seem to play a role in the susceptibility of diabetic patients to infections, particularly by resistant pathogens [19].

Infections in DFUs further aggravate the wound healing process, being responsible for frequent visits to the hospital and constituting the main complication that leads to non-traumatic amputations of lower limbs in patients with DM [20,21]. Approximately 58% of people who have a DFU will have an infection [22].

A great diversity of pathogenic and non-pathogenic microorganisms live on the skin of humans. Generally, three to five species of different microorganisms are found in an infected DFU, including the following: gram-positive aerobes (*Staphylococcus aureus*, *Staphylococcus epidermidis*, *Corynebacterium* spp.); gram-positive anaerobes (*Enterococcus* spp., *Propionibacterium* spp., *Streptococcus* spp., *Peptostreptococcus* spp., *Peptococcus* spp.); gram negative aerobes (*Pseudomonas aeruginosa*, *Acinetobacter* spp.); gram negative anaerobes (*Proteus mirabilis*, *Escherichia coli*, *Bacteroides* spp.); and fungi (*Candida* spp.) [23,24]. In low-income countries, there is a higher prevalence of gram-negative pathogens, the most common of which is *Pseudomonas aeruginos* [25,26].

## 3. Wound Healing Process in Diabetes Mellitus

A major problem with diabetic wounds is that they do not follow the normal process of wound healing, that is, the dynamic process comprising four phases: hemostasis, inflammation, proliferation, and remodeling (Figure 1).

### 3.1. Hemostasis

The first phase of the cell repair process involves platelet activation, aggregation, and adhesion to the damaged endothelium to maintain hemostasis, a phenomenon known as coagulation. Once this process is initiated, fibrinogen becomes fibrin, forming the thrombus and the temporary extracellular matrix (ECM). Other cells, such as activated platelets, neutrophils, and monocytes, which release some proteins and various growth factors, such as platelet-derived growth factor (PDGF) and transforming growth factor β (TGF-β), also participate [27]; see Figure 1a. Compared with normal subjects, hypercoagulability and a decrease in fibrinolysis are some of the changes in the hemostasis phase that have been observed in patients with DM [28].

### 3.2. Inflammation

An inflammatory process take place when a tissue injury occurs, because the neutrophils, macrophages, and mast cells are responsible for producing inflammatory cytokines, such as interleukin 1 (IL-1), interleukin 6 (IL-6), tumor necrosis factor-alpha (TNF-α), and interferon gamma (IFN-γ), as well as several growth factors, such as PDGF, epidermal growth factor (EGF), and insulin-like growth factor 1 (IGF-1), which are fundamental in the wound repair process [29,30]. In patients with DM, there exists a disequilibrium of these cytokines that leads to a modification of wound repair [31]. It has been reported that neutrophils present an altered cytokine release pattern and show a decrease in their functionality, and thus contribute to the susceptibility to wound infection [32].

### 3.3. Proliferation and Migration

When inflammation decreases, several processes start at the site of the lesion: wound contraction occurs; angiogenesis takes place to restore the oxygen supply; and ECM proteins form, including collagens, fibronectin, and vitronectin, which are necessary for cell movement, in addition to the migration of keratinocytes. All these processes are necessary for the tissue to recover its integrity and functionality [33].

Owing to hyperglycemia, the migration of fibroblasts and keratinocytes, as well as their proliferative capacity, is diminished in patients with DM. Abnormal cell migration causes a deficient re-epithelialization of the diabetic wound, affecting the healing process [27,34]. In addition, in DM patients, a decrease in angiogenesis and, therefore, a decrease in blood flow, have also been reported [35]; see Figure 1b.

### 3.4. Remodeling Phase

This phase starts approximately one week after wound healing and can last more than six months. Here, collagen that is synthesized is greater than that which is degrading and replaces the provisional ECM that was initially formed by fibrin and fibronectin. This granulation tissue becomes mature scar tissue and also increases the wound resistance, ending in the formation of a scar [36].

The fibroblasts of patients with DM are altered in their function, which contributes to defective closure of the wound; although the mechanism is not well known, it is believed that it is because of the fact that they do not respond to the action of TGF-β, as well as the aberrant production of the ECM [37].

## 4. Treatments for Diabetic Foot Ulcers

One strategy for the management of patients with a DFU is to introduce a multidisciplinary approach and address the multifactorial processes involved in DFUs. The use of multi-disciplinary teams (MDTs) that include all relevant specialties (i.e., nursing, orthopedics, plastic surgery, vascular surgery, nutrition, and endocrinology departments) has shown an effect of decreasing the risks associated with DFUs and amputation by 50–85%, lowering costs, and leading to a better quality of life for patients with DFUs [38].

Management of DFUs requires the correct classification of stage and severity. Adequate care for DFUs should include a focus on DM control as well as on wound care, proper infection control, relieving pressure, and optimizing blood flow [39]. The basic care for the control and treatment of DFUs is focused on the management of adequate perfusion, pressure mitigation, control of infection, and debridement [40]. With technological advancement, other series of therapies for DFUs have been implemented, such as the development of skin substitutes, negative pressure wound therapy, hyperbaric oxygen, the creation of new wound dressings that include growth factors, and the use of tissues from bioengineering. In this way, treatments for DPN, PAD, and infections have provided encouraging results [39,41].

### 4.1. Treatments for Diabetic Peripheral Neuropathy

DPN is a factor that increases the risk of the appearance of a DFU, owing to the loss of sensation in the limb, making patients vulnerable to trauma [42]. Tight glycemic control is the primary step and a main feature of DPN management [43]. Normoglycemia is more effectively restored by a pancreas transplant [44]. Several studies have demonstrated improvement of motor and sensory neuropathy in patients with DPN after they were treated with a pancreas transplant. However, discrepancies exist in the time of response [43,44].

Pharmacological treatment is used for painful DPN manifested as numbness, burning, stabbing, or excruciating or intractable pain; only three treatments are approved by the U.S. Food and Drug Administration for the pain associated with DPN, namely, pregabalin, tapentadol, and duloxetine [45].

Another pharmacological therapy includes analgesics, such as tramadol, acetaminophen, and some opioids such as oxycodone, which have constipation and nausea as side effects, and must be taken with care because they can be misused [46]. Therapy with antidepressants such as amitriptyline, nortriptyline, and venlafaxine, among others, has shown an efficacy in neuropathic pain management. It has an effect on the recapture of noradrenaline and serotonin, as well as on muscarinic effects. In spite of this, there are limited studies evaluating these drugs because their doses in clinical trials are not entirely reproducible in clinical practice [47].

Alpha-lipoic acid (ALA) has been suggested as a potential therapeutic agent in treating DPN; its antioxidant capacity seems to delay or reverse damages to peripheral nerves. Several human randomized controlled trials (RCTs) have investigated the effects of ALA in the development of diabetic nephropathy. A meta-analysis of four RCTs (n = 653) showed that, compared with placebo, intravenous ALA (600 mg per day) decreased symptoms of neuropathy when administered for three weeks, but symptom improvement with oral ALA (>600 mg per day for 3–5 weeks) was not clinically significant. There is no evidence evaluating long-term treatment [48].

Currently, treatments based on the use of mesenchymal stem cells (MSC) derived from adipose tissue have been considered as a potential treatment against DPN. These therapies promote the production of pro-angiogenic, neuroprotective, and anti-inflammatory factors, which have a positive impact on the clinical manifestations of the disease [49].

On the other hand, the use of biological therapy with low doses of IL-6 has been demonstrated to promote improvement of blood flow, decrease chronic inflammation, and regenerate peripheral nerve fibers. Accordingly, IL-6 may prove to be an effective treatment for the protection and/or restoration of peripheral nerve function in DPN [50]; see Table 1.

### 4.2. Treatments for Peripheral Arterial Disease (Ischemia)

Ischemia presenting in diabetic patients due to a reduction in blood flow that occurs in both small vessels (microvascular, such as capillaries) and large vessels (macrovascular, such as arteries and veins), or due to a decrease of angiogenesis, can be treated through revascularization of at least one of the foot arteries to try to restore blood flow in patients with a DFU whose toe pressure is <30 mmHg or transcutaneous oxygen pressure (TcPO2) <25 mmHg, as well as in those who have a DFU that does not heal with pressure on the ankle of <50 mmHg or ankle-brachial index (ABI) <0.5 [75].

The revascularization techniques used as a first-line strategy are open bypass or endovascular techniques [76]. Bypass is usually more effective and ensures long patency in obstruction of the common femoral artery and its bifurcation or in the case of long occlusion of femoral–popliteal and infrapopliteal vessels; however, PAD may be treated by the endovascular approach in centers with a long experience of angioplasty [77]. Alternative techniques in the treatment of PAD are angioplasty in the lower extremities, in which a small balloon is passed into a narrow section of an artery, and inflated to open up the artery in order to improve blood flow. To be effective, this procedure must be performed on a permeable distal vessel [41,78]. Atherectomy is another technique, in which the atheroma is excised by a rotating cutting blade. However, there is no evidence for superiority of atherectomy over angioplasty on any outcome [77].

Once any of the aforementioned procedures is performed, the patient must receive multidisciplinary care to make the treatment effective. This includes pharmacological treatment for hypertension, hypercholesterolemia, and bleeding as a complication [79].

### 4.3. Wound Dressings

Clinical practice guidelines recommend the use of wound dressings to maintain a humid environment in the wound, help the absorption of exudate, prevent infections, and promote the healing of ulcers [80]. The standard dressings used in diabetic wounds are those that are not adherent, such as bandages [81]. However, specialized dressings are being developed, including hydrogels [82]. Zhao et al. reported that hydrogel with insulin and fibroblasts as bioactive dressings have promising potential in the treatment of DFUs, favoring neovascularization, collagen deposition, and wound healing [83]. Regarding hydrocolloid and foam dressings, no evidence has been found to suggest that these dressings are more effective than standard dressings [84].

### 4.4. Debridement

Debridement is a procedure where the necrotic tissue that is devitalized and the surrounding border of the DFU are removed and the healthy tissue is preserved [29], and is recommended at one to four weeks, depending on the progress of healing [80]. Among its objectives, control of the bacterial load allows early closure of the wound through conservative treatment or a skin graft.

Pressure over bone prominences can lead to callus formation, which represents a foreign body that can elevate the plantar pressure and predispose subcutaneous bleeding, causing the skin to break down and ulcer formation. Debridement of calluses often results in a significant reduction in foot pressure, facilitating accurate assessment and helping close the lesion [85]. In addition, debridement can allow health personnel to carry out an evaluation of the size, shape, and depth of the wound, as well as its characteristics for a better individualized treatment [80,86]. The appropriate technique for eliminating necrotic tissue is still a matter of debate [87].

### 4.5. Relief of Pressure

The constant and repetitive traumatism of the foot owing to inadequate footwear contributes to the development of a DFU [88]. For the relief of pressure, half-shoes, rigid-soled post-operative shoes, accommodative dressings, and total contact casts, either removable or immovable, are generally used in patients with ulcers of neuropathic origin. Their use has an impact factor in the relief of plantar pressure. Each case is individually assessed for classification and determination of the necessary time to produce ulcer healing; for example, in uncomplicated plantar ulcers, a period of 32–52 days is necessary to achieve a good result. The total contact cast has been shown to be 70% effective, and is it also used as the reference treatment by the American Diabetes Association [4]. Conventional or standard therapeutic footwear is not effective in healing the ulcer, owing to poor patient adherence to recommendations for using a removable device [89]; however, they are recommended because of their low cost and the possibilities they offer to generate an adequate discharge without causing alterations of greater severity in the ulcer [90]. They are contraindicated in patients with infection or osteomyelitis [91].

### 4.6. Infection Treatment

Antibiotic therapy for infected DFU will be empirical in the first instance, in accordance with the likely causative pathogen and the severity of the infection. The definitive treatment is modified according to the results obtained in the microbiological culture and the response of the empirical treatment. Its duration will depend on the severity of the infection; for example, a mild infection could remain for 1–2 weeks, or 2–4 weeks for a severe infection, or longer if there is osteomyelitis [23].

The recommended empiric antibiotic therapy according to severity is dicloxacillin, cephalexin, clindamycin, or amoxicillin/clavulanate for mild-moderate cases; vancomycin + ampicillin/sulbactam, moxifloxacin, cefoxitin, or cefotetan for moderate cases; and vancomycin + piperacillin/tazobactam, imipenem/cilastatin, meropenem, or doripenem for severe cases [92].

Several variables present in the infection of the wound, such as time of evolution, hygiene conditions, immune status of the host, polymicrobial infection, and previous treatment with antimicrobial agents, which can collaborate with the emergence of antimicrobial resistance [93]. Antimicrobial resistance may result in prolonged debility of the patient, changing the panorama and/or the index of the wound cure [94]. The most worrisome problem for infected DFUs is gram-negative organisms that produce beta-lactamase or carbapenems and even cause resistance to methylcycline with intermediate resistance to vancomycin [95].

### 4.7. Antimicrobial Peptides

Currently, the emergence and spread of bacteria resistant to conventional antibiotics is becoming a global threat. In this sense, alternative compounds are urgently needed. Antimicrobial peptides (AMPs) are molecules of the immune system of mammals whose function is to fight the invading pathogens [96]. AMPs are potent agents against a wide spectrum of pathogens, including viruses, fungi, and antibiotic-resistant bacteria, and have antitumor activities with a complex mechanism of action. They are able to target the cytoplasmic membrane and interfere with DNA and protein synthesis, protein folding, and cell wall synthesis, thus intervening in the immunomodulatory functions, including the inflammatory process and cicatrization [73]. Mammalian antimicrobial peptides can be found in the granules of neutrophils, epithelial cells of the skin, and mucous membranes, as well as in protein degradation products [97]. Their use as monotherapy in the treatment of infections, in combination with conventional antibiotics for synergistic purposes, immunomodulators, and neutralizing endotoxins, has been suggested [73]. In DFUs, several antimicrobial peptides have been assessed including nisin, α-helical antimicrobial decapeptide KKVVFWVKFK (KSL-W), ubiquicidin 29-41 (UBI 29-41), pexiganan (MSI-78), and beta-defensin-2 (hBD2); however, only pexiganan (MSI-78) is in clinical phase development as a topical cream [98,99,100,101,102,103]. There are currently no peptides approved for use in humans by the Food and Drug Administration (FDA).

### 4.8. Larval Therapy

Larval therapy was introduced in 1940 in the United States and later abandoned after the appearance of antibiotics; however, since bacterial resistance was found, the therapy was reintroduced by the United Kingdom in the 1990s. It is still a little-used therapy, but it seems to be a promising technique in the treatment of hard-to-heal wounds [104]. The most-used maggots for the treatment of DFUs are *Lucilia cuprina*, a species of blow fly. The maggots are sterilized and placed directly on necrotic chronic wounds for evaluation of their efficacy of debridement. This procedure has been tested in diabetic patients and in different animal models of DFUs. The results showed that after treatment with several cycles of maggots, the wound was completely free of bacterial contamination and healed, and new tissues to close the wound were formed [105,106].

### 4.9. Laser Therapy

Low-level laser therapy (LLLT) involves the use of light in the form of light emitting diodes of a low-level. This treatment is considered an effective therapeutic method in wound healing when certain factors are properly observed, such as power input, dosage, time, and interval between sessions. LLLT alters the cell function, and molecular and biochemical pathways, which may result in changes in cell shape, migration, and cell signaling. These changes promote the reduction of the inflammatory phase, favoring the angiogenesis and the production of extracellular matrix components, accelerating the healing process [107]. LLLT has the advantage of being easily administered, but is considered an emerging modality of high-cost treatment with limited results, based on previous studies [108].

## 5. Tissue Engineering Approaches

Tissue engineering is a field of regenerative medicine that refers to the practice of combining scaffolds, cells, and growth factors with the goal of restoring, maintaining, or improving damaged tissues or whole organs [109]. In the case of DFUs, new dressings designed with living tissues have been developed to function as a substitute for injured skin. It is believed that their mechanism of action is to fill the wound with cells and extracellular matrix, thus inducing the expression of growth factors and cytokines that contribute to wound healing [110].

### 5.1. Growth Factors

Growth factors are proteins that can stimulate and activate the process of cell proliferation through activating angiogenesis, the transcription of genes, and other reactions that favor the closure of wounds [111]. Experimental studies have helped to elucidate the function and effectiveness of individual cytokines and growth factors in DFU tissue repair. The most common include the following: epidermal growth factor (EGF), which acts in the stimulation of epidermal cells and increases the proliferation of fibroblasts to thereby increase the production of collagen at the site of the lesion [112,113]; platelet-derived growth factor (PDGF), which stimulates fibroblasts for the formation of epithelial tissue, although no effect has been found in keratinocytes [112]; fibroblast growth factor (FGF) and transforming growth factor beta (TGF-β), which are inducers of angiogenesis; vascular endothelial growth factor (VEGF), a potent mitogen for vascular endothelial cells that can also stimulate angiogenesis; and autologous platelet-rich plasma (PRP). These growth factors were assessed parenterally through their application (alone or loaded in a dressing matrix) on the surface of the ulcer, or infiltrated directly into the wound. They contribute to the healing of wounds caused by DFUs in DM type 1 and type 2 [114,115,116]. Regarding the use of growth factors in clinical trials, only EGF has been tested in patients with DFUs, showing high cure rates [117]. The use of the other growth factors is still not recommended owing to a lack of scientific evidence on the safety profile of the damage and benefits of their use.

### 5.2. Cells

Among the cell types used for wound repair are mesenchymal stem cells (MSCs), which possess the potential for multilineage differentiation into neurogenic, chondrogenic, adipogenic, osteogenic, myogenic, and endothelial cells in the presence of lineage-specific induction factors; they are relatively easy to obtain from different tissue sources and expand in culture [118]. Several studies have evaluated the regenerative potential of MSCs derived from bone marrow and umbilical cord blood, by administrating them intramuscularly, intra-arterially, and topically in diabetic patients with DFUs in clinical trials. The results showed that they can accelerate wound closure, ameliorate clinical parameters, and improve the healing of DFUs [119]. In spite of the promising results in preclinical studies, as well as their safety and efficacy in the treatment of DFUs, the appropriate cell type and selection between autologous or allogeneic MSCs are yet to be discussed. In the same way, MSCs derived from adipose tissue were investigated in combination with Exendin-4 to evaluate their improvement effect on diabetic wound healing in a diabetic murine model. The results showed a rapid reduction in the size of the wound compared with controls [120]. A mechanism proposed for the healing effect of MSCs is their paracrine effect over both cell proliferation and the process of angiogenesis, as well as the regulation of local mRNA expression of some factors involved in healing [121].

In the same sense, fibroblasts and keratinocytes were evaluated alone or in combination as allografts or autografts to cover chronic ulcers for treatment of DFUs [122]. These cells are known to enhance wound healing by synthesizing several growth factors and major extracellular matrix components of the dermal and epidermal layers of the skin [123].

### 5.3. Scaffolds

The biomaterials currently used as matrices for wound repair can be classified according to their origin, either natural or synthetic. Among the natural matrices are collagen [56], hyaluronic acid [124], fibrin [125], chitosan [126], and alginate [127], which are recommended because of their easy degradation and high biocompatibility. Among the synthetic matrices are poly(acrylic acid) (PAA) [128], polyglycolic acid (PGA) [129], poly(lactic-co-glycolic acid) (PLGA) [130], poly(e-caprolactone) (PCL) [131], PCL-poly(ethylene glycol) (PEG) [132], gelatin methacrylate (GelMA) [133], and pluronic F-127 [134], which are recommended to give resistance to scaffolding material. The biomaterials mentioned above are used as hydrogels, bandages, foam, and films.

### 5.4. Human Skin Substitutes

The use of skin substitutes that combine growth factors, cells, and/or biomaterials as a treatment for DFUs has been widely suggested and accepted. There are a great variety of skin substitutes made through tissue engineering. The mechanism by which they help wound healing depends on their anatomical shape and the main components and biomaterials. Examples include the following: Dermagraf^®^, which owes its effect to its components such as metabolically active fibroblasts, components of the ECM, and a bioabsorbable synthetic scaffold [135]; Apligraf^®^, whose healing effect is the result of the production of cytokines and growth factors similar to healthy human skin [136]; Regranex^®^ (Becaplermin), which has biological activity similar to natural PDGF, and which has been shown to promote the formation of granulation tissue and aid wound healing [137]; and AlloDerm^®^, which consists of an organized acellular dermal matrix of normal collagen, allowing new human skin to regenerate and causing neovascularization [138]. The main products of regenerative medicine available and their characteristics are summarized in Table 2.

All these products help repair the skin and their main advantages include the ability to be supplemented with growth factors and drugs, such as antibiotics and anti-inflammatories, in order to achieve adequate wound healing. However, the cost associated with some devices must be considered [150]. These products act on different layers of the skin because they are developed based on different principles, origin, and anatomy; Figure 2 shows how these products are constituted.

## 6. Conclusions

DFUs precede 85% of lower extremity amputations. They often lead to complications such as infection, osteomyelitis, and abscesses. DFUs greatly impact physical, psychological, social, and economic aspects for patients, affecting their quality of life. Management of DFUs requires the correct classification of stage and severity. Adequate care for DFUs should include a focus on DM control as well as on wound care, proper infection control, relieving pressure, and optimizing blood flow. The basic care for the control and treatment of DFUs includes discharge, debridement processes, revascularization, and antibiotic therapy. Therapeutic footwear reduces ulcer recurrence, although few studies are available that have assessed its efficacy and the prevention of initial ulceration. Several methods of debridement (including the use of larvae) and revascularization have been used to improve the healing process of ischemic ulcers. They have been shown to be effective in reducing pain and increasing arterial flow to the ischemic limb, as well as reducing the risk of amputation. However, more studies are needed to determine the patient populations for which these therapies are helpful, as well as their cost-to-benefit ratios. AMPs are potent agents against a wide spectrum of pathogens, including viruses, fungi, and antibiotic-resistant bacteria, and have antitumor activity, which represents an alternative treatment to conventional antibiotic therapy.

With technological advancement, a number of biological healing products have been developed to aid in the healing process of DFUs. Tissue engineering represents a novel treatment for DFUs, including the use of dressings for wounds designed with living tissues to function as skin substitutes. These skin substitutes use living cells such as fibroblasts, keratinocytes, and stem cells, alone or in combination with extracellular matrices, and growth factors. Even though the use of these new therapies is expensive, they are supported by several clinical trials, encouraging their use, and representing potential new therapies for the future.

## Figures and Tables

**Figure 1 medicina-55-00714-f001:**
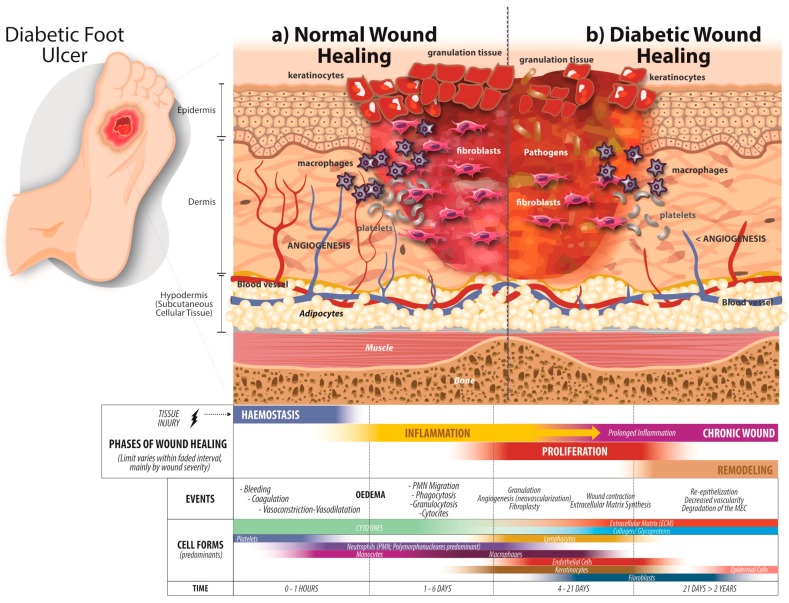
Wound healing process in diabetes mellitus. (**a**) Normal wound healing. In healthy people, wound closure consists of several processes that occur sequentially: the rapid hemostasis that involves platelet aggregation to form the platelet plug; an inflammation phase where neutrophils, macrophages, and mast cells release proinflammatory cytokines; wound contraction when inflammation decreases, angiogenesis occurs, keratinocytes and fibroblasts migrate, and the extracellular matrix forms; and, finally, the remodeling phase, where granulation tissue converts into mature scar tissue. (**b**) Diabetic wound healing. In patients with diabetes mellitus (DM), the wound closure processes are affected, starting with a decrease in fibrinolysis and an imbalance of cytokines, which causes an alteration in wound closure. There is also a decrease in angiogenesis due to hyperglycemia, and the migration of cells such as keratinocytes and fibroblasts is diminished, causing deficient re-epithelialization; in the same way, the poor production of the extracellular matrix (ECM) by fibroblasts contributes to the problem of a deficient wound closure.

**Figure 2 medicina-55-00714-f002:**
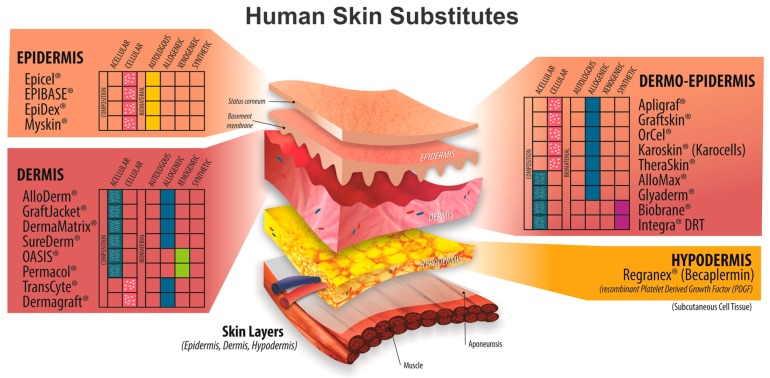
Human skin substitutes. Regenerative medicine products are based on different principles and components: some are integrated with cells, such as Dermagraf^®^, Epicel^®^, EpiDex^®^, EPIBASE, Myskin^®^, TransCyte^®^, Apligraf^®^, Graftskin, OrCel^®^, Karoskin^®^, and TheraSkin ^®^; others are acellular, such as AlloDerm^®^, GraftJacket^®^, DermaMatrix^®^, OASIS^®^, SureDerm^®^, Permacol^®^, AlloMax^®^, Glyaderm^®^, Biobrane^®^, and Integra^®^. The figure also outlines how these products have different anatomical origins, including the epidermis, such as Epicel^®^, EpiDex^®^, EPIBASE, and Myskin^®^; the dermis, such as AlloDerm^®^, GraftJacket^®^, DermaMatrix^®^, OASIS^®^, SureDerm^®^, Permacol^®^, TransCyte^®^, and Dermagraf^®^; dermo-epidemics, such as Apligraf^®^, Graftskin, OrCel^®^, Karoskin^®^, TheraSkin^®^, AlloMax^®^, Glyaderm^®^, Biobrane^®^, and Integra^®^; and the hypodermis, such as Regranex^®^ (Becaplermin).

**Table 1 medicina-55-00714-t001:** Therapeutic strategies for diabetic foot ulcer (DFU) management.

Therapy for	Method	Advantage	Disadvantage	Reference
**Neuropathic Ulcer**	Anticonvulsants: Gabapentin, Pregabalin	Neuropathic pain reduction.	Dyspnea, drowsiness, fatigue. The effect occurs after the second week.	[51,52]
	Antidepressants: Duloxetine, Amitriptyline, Nortriptyline, Venlafaxine	Good effect against the neuropathic pain. Effects similar to gabapentin and pregabalin.	Sleep disturbances, depression, and have muscarinic effects.	[46,47]
	Analgesics: Tapentadol, Tramadol, Acetaminophen, Oxycodone	Reduce pain in diabetic polyneuropathy.	Confusion and sedation; opioids can be used inappropriately.	[11,46]
	Alpha-lipoic acid	Delay or reverse damages to peripheral nerves.	There is no evidence evaluating long-term treatment.	[48]
	Mesenchymal stem cells	Neuroprotective effects. It can be easily isolated from adipose tissue; has cell plasticity.	The number of transplanted cells that reach and are integrated into the functioning of the organ is low. The therapies are expensive.	[49]
	Interleukin 6	Regenerates peripheral nerve fibers.	High doses can cause inflammation.	[50]
**Ischemic Ulcer:(a) Endovascular therapy**	Angiosomas	Increases arterial flow to the ischemic limb. Get at least one pulsatile flow. Improves the healing of ischemic ulcer. Improves or eliminates pain at rest Reduce the level of amputation. Reduce the duration and number of hospitalizations. Improve mobility. Improves quality of life. Improves survival.	Variability in infrapopliteal arterial distribution. Differences between extension and borders of angiosomes. Difficulties in identifying affected angiosoma. Many lesions depend on several angiosomes. Objective diagnostic angiographic pattern not described. Optimal angiographic end point post endovascular therapy is not known. Differences in collateralization. Very long arterial segments. Diffuse, calcified, and multiple lesions. Small arterial caliber. Slow flow of distal beds. Poor run-off. Instrument handling. Technical difficulties.	[53,54,55]
	Percutaneous transluminal angioplasty	Technical feasibility reduces the number of complications, and increases the rate of recovery of the limb useful in elderly patient.	Limited scientific evidence. Not suitable for young patients. Requires adjuvant treatment to prevent restenosis with platelet inhibitors or vitamin antagonists K.	[55,56]
	Stents	Improves blood flow.	The permeability of the arteries after an angioplasty is the same if this is placed than if it is omitted.	[57,58]
	Angioplasty	Increases the primary permeability of the vessel. Revascularization of the target lesion.	High percentage of restenosis. Does not decrease the risk of amputation High cost.	[59]
	Bypass: autologous human umbilical vein and synthetic materials with or without heparin	Improve primary permeability. Preservation of the foot.	Lack of scientific evidence.	[60,61]
**(b) Anticoagulant Therapy **	Plaquetary inhibitors. Antagonists of vitamin K.	Adjuvant after angioplasty.	Hemorrhages. Hypersensitivity. Gastrointestinal disorders.	[62]
	Ginkgo biloba	Improves intermittent claudication.	Lack of scientific evidence.	[63]
	Vitamin E	Improves blood flow. Increases the body’s ability to repair. No adverse effects. Low cost.	Lack of documented scientific evidence.	[64]
	Levocarnitine	Improves walking tolerance. Greater effectiveness intravenously. Severe claudication better results.	There are not enough studies documenting their effectiveness in these patients. No dose has been established, and duration of treatment for patient safety.	[65,66]
	Beta-blockers	Its use does not affect walking distance, blood flow, the vascular resistance of the leg, or skin temperature.	Lack of scientific evidence.	[67]
	Cilostazol	Improve walking distance.	Presents mild and treatable side effects. Lack of scientific evidence.	[68]
	Hyperbaric oxygen therapy/ozone	Improves symptoms. Decreases ulcer area. Shorter duration of hospitalization.	The studies found are small, and there is a high risk of bias. It requires adjuvants with antibiotics.	[69,70]
	Stimulation of the spinal cord	Decreased pain. Greater limb preservation rate. Regression of the ischemic limb state (Fontaine). Improves the quality of life.	High cost, the risk of complications, such as implantation problems, infections that will eventually require reoperation.	[71,72]
**Infection**	Antibiotics	Selective. Low cost. Mechanism of specific action. Established doses. Multiple administration routes.	Drug interactions, high resistance potential hypersensitivity.	[73]
	Antimicrobial peptides of mammals	Multiple mechanisms of action. Broad-spectrum antimicrobial. Low resistance potential. Antiviral, antifungal, antibacterial, antitumor activity.	Its toxicity is unknown; it can only be administered topically. Embryotoxic and paralyzing activity for sperm. Short half-life, for the degradation of proteases, high price.	[73,74]

**Table 2 medicina-55-00714-t002:** Description of human skin substitutes for the treatment of DFUs. FDA, Food and Drug Administration.

Type of Product	Name/Clinical Phase	Composition
Cell therapy	Dermagraft/Approved by the FDA.	Dermal substitute derived from cryopreserved human fibroblasts composed of fibroblasts, extracellular matrix, and a bioabsorbable scaffold [139].
Cell therapy	Apligraf/Approved by the FDA in 1998.	Two-layer skin substitute: the epidermal layer is composed of human keratinocytes; the dermal layer is formed by human fibroblasts in a type I bovine collagen matrix [140].
Cell therapy	Becaplermin/Approved by the FDA.	Transparent colorless to straw-colored gel, which contains 0.01% of the active ingredient becaplermin [137].
Cell therapy	OrCel/Approved by the FDA.	The cultivated skin compound is an absorbable bilayer of cellular matrix, made of bovine collagen, in which the dermal cells have been cultivated [141].
Cell therapy	Epicel/Approved by the FDA.	Autograft grew for deep or full-thickness dermal treatment comprising a surface area of greater than 30% [142].
Biosynthetic	Biobrane/Approved by the FDA.	Biosynthetic dressing for wounds, consisting of a single silicon film with a nylon fabric partially embedded in the film. The fabric creates a complex of the three-dimensional structure of thread trifilamento, which chemically binds to the collagen. The blood and serum form a clot in the nylon matrix of the dressing that adheres to the wound until epithelization occurs [143].
Biosynthetic	Integra/Approved by the FDA.	A compound of bovine collagen with dermal glycosaminoglycans coated with a silicone, as a temporary epidermal substitute [144].
Biosynthetic	TansCyte/Approved by the FDA in 1997.	Human dermal fibroblasts cultured in a nylon mesh, combined with a synthetic epidermal layer. Used as a temporary cover for some wounds that heal without autografting [145].
Collagen Support	OASIS/Approved by the FDA.	Support of xenogenic collagen derived from the porcine intestinal mucosa [146].
Acellular Dermal Matrix	AlloDerm/Approved by the FDA.	Acellular dermal matrix dressing (allograft), used as a replacement tissue. The product is created from the native human skin and processed so that the basement membrane and the cellular matrix remain intact [147].
Acellular Dermal Matrix	DermaMatrix/Unknown	Acellular dermal matrix (allograft) from donated human skin tissues; processed by the skeletal muscle [148].
Acellular Dermal Matrix	GraftJacket/Approved by the FDA.	Formed by a matrix of acellular regenerative tissue that has been processed from the donation of human skin; minimally processed to eliminate epidermal and dermal cells, while preserving the skin structure at the same time [149].

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
