# Peer review of "Current Therapeutic Strategies in Diabetic Foot Ulcers"

_medicina, 2019, doi:10.3390/medicina55110714_

Round 1

Reviewer 1 Report

The main objective of this manuscript was to review therapeutic approaches for Diabetic Foot Ulcer treatment particularly tissue engineering techniques. The manuscript is well-written, figures are clear and informative. The information gathered here may benefit to researchers and health care professionals in order to develop new strategies for effective Diabetic Foot Ulcer management.

Author Response

Authors appreciate the invaluable Reviewer comments.

Reviewer 2 Report

General comments

The authors need to engage an individual with solid knowledge in English. There are a high number of very long sentences which may cause the reader to loose focus and also many sentences which are difficult to understand.

In general the authors do not provide a critical approach to this review. It is merely a summarizing of potential treatment strategies and to a minor extend a critical evaluation of different studies.

Standard of care treatment constitutes a minor part of the authors suggestions – and no critical approach is put forward here.

The use of MDT´s (Multi-Disciplinary Teams) have shown an effect on amputation rates and these setups should be mentioned by the authors. A fast approach to the patient with a DFU is often mandatory and can save limbs by usage of all relevant specialties (i.e. orthopedic surgeon, plastic surgeon, vascular surgeon and endocrinologist).

The authors do not mention anything on media sclerosis, but only express the deteriorating effect of atherosclerosis. In the diabetic foot syndrome media sclerosis is very often seen (in particular in individuals with Charcot foot) – does this not deserve mentioning in the eyes of the authors? Even though it may not have a high impact on morbidity and mortality, but indeed on the ability to (technically) perform by-pass surgery.

Specific comments

Line 35: it is stated that “DFU is the major complication of diabetes”. We all know that around 15-20% (as also stated by the authors) develop DFU lifelong. For diabetic kidney disease it is probably more close to 30% - thus DFU is not the major complication of diabetes. Otherwise the authors must substantiate what they really mean.

Line 42: the authors state that DFU is preventable. That is somewhat of a postulate and should be substantiated instead of presenting it as a definite truth.

Line 57-58: “which are warning signs in the body to avoid injuries” – statements like these are self-evident and are rather disperse throughout the manuscript. Should be avoided.

Line 77: “ a reduction in the immune system activity” – this should be a bit more elaborated.

Line 158: There is no (at least not well documented) effective treatment for DPN except optimal glycemic control - Gabapentin etc. are for painful diabetic neuropathy which is a special entity! The authors confuse the reader with this statement. In general section 4.1 is short of references and mainly contains statements.

Line 167-168: here is another example of the lining up of treatments without considering the effect in a critical manner. What is the numbers needed to treat and needed to harm etc.

In table 1 (latter section) the word “Balloons” is used – I believe it would be more proper to use the expression angioplasty (e.g. balloon dilation). The table is rather exhaustive – yet there is no mention of Alfa-lipoic acid, which has been examined rather extensively (e.g. the SYDNEY trial a.o.)

Line 180: ischemia is probably not only reserved for macrovascular disease, but also microvascular disease. Often ulcers are formed in a diabetic foot with peripheral artery disease (i.e. small arteries).

Line 185 ff: to much unnecessary information is put in place here. The authors should be much more stringent in their information and not put self-evident information into a review like this.

Line 205-207: pure nonsense in my opinion!

Line 213 ff: it seems that the authors do not think that callus removal is essential. I strongly disagree as callus formation would increase the risk of subcutaneous bleeding and ulcer formation. Again, speculative, even though the authors find one reference to substantiate their statement.

Line 240-245: one long sentence! You are sure to loose your reader by writing extremely long sentences!

Line 247: dicloxacillin is considered less useful for G+ organisms (e.g. Staphylococcus aur.). This is not in agreement with the antibiotic strategy used in several countries. Using Levofloxacin is tied to a much higher risk of developing bacterial resistance. The authors should speculate on this in a more critical manner. Long-term therapy (as in osteomyelitis) with f.i. Levofloxacin is not a proper solution.

Line 257: a peculiar start of the sentence – must be a mistake!?

Line 274: usually the literature use “hard-to-heal” wounds

Line 308-309: the authors should involve a person with proper knowledge in English to ascertain a correct formulation of this sentence.

Line 380-383: Again here is something self-evident. The conclusion should be much more stringent. In line 383 it says: “Most DFUs are treated in a conventional  manner, which includes discharge,…” this sentence is by no means clear!

Author Response

Reviewer(s)' Comments to Author:

Reviewer: 2

Comments and Suggestions for Authors

General comments

The authors need to engage an individual with solid knowledge in English. There are a high number of very long sentences which may cause the reader to loose focus and also many sentences which are difficult to understand.

Response: To attend this issue, we requested the MDPI English Editing Service. At the final of this document, I put the english-editing-certificate of the manuscript.

In general the authors do not provide a critical approach to this review. It is merely a summarizing of potential treatment strategies and to a minor extend a critical evaluation of different studies.

Standard of care treatment constitutes a minor part of the authors suggestions – and no critical approach is put forward here.

Response: We are really sorry for the reviewer appreciation. To attend this issue, we tried to do our best correcting the manuscript with the specific comments of the reviewers.

The use of MDT´s (Multi-Disciplinary Teams) have shown an effect on amputation rates and these setups should be mentioned by the authors. A fast approach to the patient with a DFU is often mandatory and can save limbs by usage of all relevant specialties (i.e. orthopedic surgeon, plastic surgeon, vascular surgeon and endocrinologist).

Response: We add the information suggested by the reviewer in the 4. Treatments for DFU section (page 4, lines 157 – 162), as follows: “One strategy for the management of patients with a DFU is to introduce a multidisciplinary approach and address the multifactorial processes involved in DFUs. The use of multi-disciplinary teams (MDTs) that include all relevant specialties (i.e., nursing, orthopedics, plastic surgery, vascular surgery, nutrition, and endocrinology departments) have shown an effect of decreasing the risks associated with DFUs and amputation by 50%–85 %, lowering costs, and leading to a better quality of life for patients with DFUs [38]”.

The authors do not mention anything on media sclerosis, but only express the deteriorating effect of atherosclerosis. In the diabetic foot syndrome media sclerosis is very often seen (in particular in individuals with Charcot foot) – does this not deserve mentioning in the eyes of the authors? Even though it may not have a high impact on morbidity and mortality, but indeed on the ability to (technically) perform by-pass surgery.

Response: We add the information suggested by the reviewer in the 2.2. Peripheral arterial disease section (page 2, lines 72 – 75), as follows: “Related to atherosclerosis is media sclerosis Mönckeberg (MSM), which is frequent in type 2 DM, and is associated with progressive calcifications of the arterial wall media leaving the intima intact [15]. MSM may interfere with compensatory remodeling of the arterial walls, and may accelerate the stenotic phase of the atherosclerosis [15, 16]”.

Specific comments

Line 35: it is stated that “DFU is the major complication of diabetes”. We all know that around 15-20% (as also stated by the authors) develop DFU lifelong. For diabetic kidney disease it is probably more close to 30% - thus DFU is not the major complication of diabetes. Otherwise the authors must substantiate what they really mean.

Response: The reviewer comment is pertinent. To attend this issue, we edit the paragraph (page 1, lines 36 – 39), as follows: “Chronic complications of the disease vary according to the type of diabetes, time since onset and degree of metabolic control, the most prevalent being: neuropathy, 25%; retinopathy, 32%; and nephropathy, 23% [3]. The diabetic foot ulcer (DFU) is a complication of DM, with an annual incidence of 2.4%–2.6% and a prevalence of 4%–10% [4]”.

Line 42: the authors state that DFU is preventable. That is somewhat of a postulate and should be substantiated instead of presenting it as a definite truth.

Response: The reviewer comment is pertinent. To attend this issue, we edit the paragraph (page 1, lines 44 – 45), as follows: “Educating patients about podiatric care could prevent DFU development, and reduce their incidence”.

Line 57-58: “which are warning signs in the body to avoid injuries” – statements like these are self-evident and are rather disperse throughout the manuscript. Should be avoided.

Response: The reviewer comment is pertinent. The sentence “which are warning signs in the body to avoid injuries” was deleted.

Line 77: “a reduction in the immune system activity” – this should be a bit more elaborated.

Response: The reviewer comment is pertinent. To attend this issue, we edit the paragraph (page 2, lines 80 – 88), as follows: “pathogen-related factors such as density, virulence and interactions, and immune defects linked to the host [9]. Several immunological defects have been reported in patients with DM, such as altered phagocytosis and bactericidal activity of polymorphonuclear cells; impaired chemotaxis and phagocytosis functions of monocyte/macrophage; disturbances of cellular innate immunity, including a low serum level of complement factor 4 (C4) and abnormal production of cytokines by monocytes [17]; and alteration of lymphocytes subpopulations and immunoglobulins levels [18]. These abnormalities, mostly related to innate immunity, seem to play a role in the susceptibility of diabetic patients to infections, particularly by resistant pathogens [19]”.

Line 158: There is no (at least not well documented) effective treatment for DPN except optimal glycemic control - Gabapentin etc. are for painful diabetic neuropathy which is a special entity! The authors confuse the reader with this statement. In general section 4.1 is short of references and mainly contains statements.

Response: The reviewer comment is pertinent. To attend this issue, we rewrite the section 4.1 Treatments for DPN (page 5, lines 175 – 190), as follows: “Tight glycemic control is the primary step and a main feature of DPN management [43]. Normoglycemia is more effectively restored by a pancreas transplant [44]. Several studies have demonstrated improvement of motor and sensory neuropathy in patients with DPN after they were treated with a pancreas transplant. However, discrepancies exist in the time of response [43, 44].

Pharmacological treatment is used for painful DPN manifested as numbness, burning, stabbing, or excruciating or intractable pain; only three treatments are approved by the US Food and Drug Administration for the pain associated with DPN, namely, pregabalin, tapentadol, and duloxetine [45].

Another pharmacological therapy includes analgesics, such as tramadol, acetaminophen and some opioids such as oxycodone, which have as side effects constipation and nausea, and must be taken with care because they can be misused [46]. Therapy with antidepressants such as amitriptyline, nortriptyline, venlafaxine, among others, has shown an efficacy in neuropathic pain management. It has an effect on the recapture of noradrenaline and serotonin, as well as on muscarinic effects. In spite of this, there are limited studies evaluating these drugs because their doses in clinical trials are not entirely reproducible in clinical practice [47]”.

Line 167-168: here is another example of the lining up of treatments without considering the effect in a critical manner. What is the numbers needed to treat and needed to harm etc.

Response: The reviewer comment is pertinent. To attend this issue, we rewrite the section 4.1 Treatments for DPN (page 5, lines 175 – 190), as follows: “Tight glycemic control is the primary step and a main feature of DPN management [43]. Normoglycemia is more effectively restored by a pancreas transplant [44]. Several studies have demonstrated improvement of motor and sensory neuropathy in patients with DPN after they were treated with a pancreas transplant. However, discrepancies exist in the time of response [43, 44].

Pharmacological treatment is used for painful DPN manifested as numbness, burning, stabbing, or excruciating or intractable pain; only three treatments are approved by the US Food and Drug Administration for the pain associated with DPN, namely, pregabalin, tapentadol, and duloxetine [45].

Another pharmacological therapy includes analgesics, such as tramadol, acetaminophen and some opioids such as oxycodone, which have as side effects constipation and nausea, and must be taken with care because they can be misused [46]. Therapy with antidepressants such as amitriptyline, nortriptyline, venlafaxine, among others, has shown an efficacy in neuropathic pain management. It has an effect on the recapture of noradrenaline and serotonin, as well as on muscarinic effects. In spite of this, there are limited studies evaluating these drugs because their doses in clinical trials are not entirely reproducible in clinical practice [47]”.

In table 1 (latter section) the word “Balloons” is used – I believe it would be more proper to use the expression angioplasty (e.g. balloon dilation). The table is rather exhaustive – yet there is no mention of Alfa-lipoic acid, which has been examined rather extensively (e.g. the SYDNEY trial a.o.)

Response: The reviewer comment is pertinent. To attend this issue, we edit the Table 1 (pages 6 – 8), replacing the word Balloon by Angioplasty, and adding de corresponding information for Alfa-lipolic acid in the table and in the text (page 5, lines 191 - 197), as follow: “Alpha-lipoic acid (ALA) has been suggested as a potential therapeutic agent in treating DPN; its antioxidant capacity seems to delay or reverse damages to peripheral nerves. Several human randomized controlled trials (RCTs) have investigated the effects of ALA in the development of diabetic nephropathy. A meta-analysis of four RCTs (n = 653) showed that compared with placebo, intravenous ALA (600 mg per day) decreased symptoms of neuropathy when administered for three weeks, but symptom improvement with oral ALA (>600 mg per day for 3–5 weeks) was not clinically significant. There is no evidence evaluating long-term treatment [48]”.

Line 180: ischemia is probably not only reserved for macrovascular disease, but also microvascular disease. Often ulcers are formed in a diabetic foot with peripheral artery disease (i.e. small arteries).

Response: The reviewer comment is pertinent. To attend this issue, we edit the paragraph (page 9, lines 210 – 212), as follows: “Ischemia presenting in diabetic patients due to a reduction in blood flow that occurs in both small vessels (microvascular, such as capillaries), and large vessels (macrovascular, such as arteries and veins), or due to a decrease of angiogenesis,”.

Line 185 ff: to much unnecessary information is put in place here. The authors should be much more stringent in their information and not put self-evident information into a review like this.

Answer: The reviewer comment is pertinent. To attend this issue, we rewrite the paragraph (page 9, lines 216 – 225), as follows: “The revascularization techniques used as a first-line strategy are open bypass or endovascular techniques [76]. Bypass is usually more effective and ensures long patency in obstruction of the common femoral artery and its bifurcation or in case of long occlusion of femoral–popliteal and infrapopliteal vessels; however, PAD may be treated by the endovascular approach in centers with long experience of angioplasty [77]. Alternative techniques in the treatment of PAD are angioplasty in the lower extremities, in which a small balloon is passed into a narrow section of an artery, and inflated to open up the artery in order to improve blood flow. To be effective, this procedure must be performed on a permeable distal vessel [41, 78], and atherectomy, in which the atheroma is excised by a rotating cutting blade. However, there is no evidence for superiority of atherectomy over angioplasty on any outcome [77]”.

Line 205-207: pure nonsense in my opinion!

Response: The reviewer comment is pertinent. To attend this issue, we delete the sentence.

Line 213 ff: it seems that the authors do not think that callus removal is essential. I strongly disagree as callus formation would increase the risk of subcutaneous bleeding and ulcer formation. Again, speculative, even though the authors find one reference to substantiate their statement.

Response: The reviewer comment is pertinent. To attend this issue, we edit the paragraph (page 9, lines 243 – 246), as follows: “Pressure over bone prominences can lead to callus formation, which represents a foreign body that can elevate the plantar pressure and predispose subcutaneous bleeding, the skin to break down and ulcer formation. Debridement of calluses often results in a significant reduction in foot pressure, facilitating accurate assessment and helping close the lesion [85]”.

Line 240-245: one long sentence! You are sure to loose your reader by writing extremely long sentences!

Response: The reviewer comment is pertinent. To attend this issue, we made the sentences shorter, and edit the paragraph (page 10, lines 275 – 281), as follows: “Several variables present in the infection of the wound, such as time of evolution, hygiene conditions, immune status of the host, polymicrobial infection, and previous treatment with antimicrobial agents, which can collaborate with the emergence of antimicrobial resistance [93]. Antimicrobial-resistance may result in prolonged debility of the patient, changing the panorama and/or the index of the wound cure [94]. The most worrisome problem for infected DFUs are gram-negative organisms that produce beta-lactamase or carbapenems and even cause resistance to methylcycline with intermediate resistance to vancomycin [95]”.

Line 247: dicloxacillin is considered less useful for G+ organisms (e.g. Staphylococcus aur.). This is not in agreement with the antibiotic strategy used in several countries. Using Levofloxacin is tied to a much higher risk of developing bacterial resistance. The authors should speculate on this in a more critical manner. Long-term therapy (as in osteomyelitis) with f.i. Levofloxacin is not a proper solution.

Response: The reviewer comment is pertinent. To attend this issue, we rewrite the paragraph (page 10, lines 271 – 274), as follows: “The recommended empiric antibiotic therapy according to severity is dicloxacillin, cephalexin, clindamycin, or amoxicillin/clavulanate for mild-moderate cases; vancomycin + ampicillin/sulbactam, moxifloxacin, cefoxitin, or cefotetan for moderate cases; and vancomycin + piperacillin/tazobactam, imipenem/cilastatin, meropenem, or doripenem for severe cases [92]”.

Line 257: a peculiar start of the sentence – must be a mistake!?

Response: The reviewer comment is pertinent. The mistake was corrected (page 10, line 286).

Line 274: usually the literature use “hard-to-heal” wounds

Response: The reviewer comment is pertinent. The mistake was corrected (page 11, line 303).

Line 308-309: The authors should involve a person with proper knowledge in English to ascertain a correct formulation of this sentence.

Response: The reviewer comment is pertinent. To attend this issue, we edit the paragraph (page 11, lines 337 – 339), as follows: “These growth factors have been assessed parenterally through their application (alone or loaded in a dressing matrix) on the surface of the ulcer, or infiltrated directly into the wound”, and we requested the MDPI English Editing Service for the manuscript. At the final of this document, I put the english-editing-certificate of the manuscript.

Line 380-383: Again here is something self-evident. The conclusion should be much more stringent. In line 383 it says: “Most DFUs are treated in a conventional manner, which includes discharge,…” this sentence is by no means clear!

Response: The reviewer comment is pertinent. To attend this issue, we rewrite de 6. Conclusion section (page 14, lines 404 – 425) as follows: “DFUs precede 85% of lower extremity amputations. They often lead to complications such as infection, osteomyelitis, and abscesses. DFUs greatly impact physical, psychological, social, and economic aspects for patients, affecting their quality of life. Management of DFUs requires the correct classification of stage and severity. Adequate care for DFUs should include a focus on DM control as well as on wound care, proper infection control, relieving pressure and optimizing blood flow. The basic care for the control and treatment of DFUs includes discharge, debridement processes, revascularization and antibiotic therapy. Therapeutic footwear reduces ulcer recurrence, although few studies are available that have assessed its efficacy and prevention of initial ulceration. Several methods of debridement (including the use of larvae) and revascularization have been used to improve the healing process of ischemic ulcers. They have been shown to be effective in reducing pain and increasing arterial flow to the ischemic limb, and reducing the risk of amputation. However, more studies are needed to determine the patient populations for which these therapies are helpful, and their cost-to-benefit ratios. AMPs are potent agents against a wide spectrum of pathogens, including viruses, fungi, and antibiotic-resistant bacteria, and have antitumor activity, which represents an alternative treatment to conventional antibiotic therapy.

With technological advancement, a number of biological healing products have been developed to aid in the healing process of DFUs. Tissue engineering represents a novel treatment for DFUs, including the use of dressings for wounds designed with living tissues to function as skin substitutes. These skin substitutes use living cells such as fibroblasts, keratinocytes, and stem cells, alone or in combination with extracellular matrices, and growth factors. Even though the use of these new therapies is expensive, they are supported by several clinical trials, encouraging their use, and representing potential new therapies for the future”.

Round 2

Reviewer 2 Report

I find that the manuscript has improved considerably and should be accepted for publication provided minor corrections in the text which I believe will be performed in the editorial process.